# Enhanced Age-Dependent Motor Impairment in Males of *Drosophila melanogaster* Modeling Spinocerebellar Ataxia Type 1 Is Linked to Dysregulation of a Matrix Metalloproteinase

**DOI:** 10.3390/biology13110854

**Published:** 2024-10-23

**Authors:** Emma M. Palmer, Caleb A. Snoddy, Peyton M. York, Sydney M. Davis, Madelyn F. Hunter, Natraj Krishnan

**Affiliations:** Department of Biochemistry, Molecular Biology, Entomology and Plant Pathology, Mississippi State University, Mississippi State, MS 39762, USA

**Keywords:** ataxin1, *Drosophila melanogaster*, lifespan, matrix metalloproteinase, polyglutamine

## Abstract

Polyglutamine (polyQ) diseases are neurodegenerative disorders caused by CAG trinucleotide repeat expansions in certain genes. Despite their simple genetic basis, their underlying causes remain unclear. *Drosophila melanogaster* has been a valuable model for studying these diseases, mimicking key features like protein aggregates and neuronal degeneration. In this study, expressing human Ataxin-1 with a long polyQ repeat in neurons led to shortened lifespan and worsened motor function in male flies. These changes were linked to increased matrix metalloproteinase 1 (dMMP1) and decreased extracellular matrix signaling and survival motor neuron gene expression, suggesting a role for matrix metalloproteinase dysregulation in overt motor impairment linked to polyQ disease.

## 1. Introduction

Neurodegenerative diseases (NDs) are a group of devastating disorders that impair movement and lead to cognitive decline by affecting specific neurons in the central nervous system. These diseases are characterized by the progressive loss of disease-specific populations of neurons [1]. NDs are termed “progressive”, since the pathology associated with these diseases continue to get worse with age, often leading to disability and death. Polyglutamine diseases, also known as polyQ diseases, are a type of ND which is dominantly inherited. PolyQ diseases are characterized by abnormal expansion of CAG trinucleotide repeats, specifically in protein-coding regions. CAG encodes for the amino acid glutamine, which is a polar, non-charged amino acid. In normal protein-coding regions, there are approximately 6 to 34 CAG repeats, while in mutated protein-coding regions there are more than 35 CAG repeats [2]. Excess trinucleotide repeats lead to aggregation of proteins. Spinocerebellar ataxias (SCAs) are a large complex group of inherited neurodegenerative disorders characterized by progressive cerebellar ataxia, oculomotor abnormalities, and a range of variable neurological features [3,4]. Spinocerebellar ataxia Type 1 (SCA1) is a relatively rare autosomal-dominant neurological disorder [5]. This disease has the intriguing feature that the disease-causing mutation is the expansion of an unstable trinucleotide repeat, specifically a CAG repeat that encodes the amino acid glutamine in ataxin-1. SCA1 differs from other SCAs in that the gene affected is *Atx-1* (or *ATXN-1*) [6]. The relationship between polyQ diseases and how they affect neuronal function through matrix metalloproteinases (MMPs) and extracellular matrix (ECM) factors are unclear.

MMPs are a family of zinc- and calcium-dependent endopeptidases that are responsible for degrading ECM proteins. Humans express 23 different MMPs regulated by four different types of tissue inhibitors of metalloproteinases (TIMPs) and the MMPs require proteolytic cleavage for their activation [7]. MMPs are essential for ECM remodeling [8]. The ECM can serve as a scaffold, anchoring cells and fostering an environment for growth, and is highly integrated into cell signaling pathways providing cues to migrate, grow, differentiate, senesce or, in neurodegenerative diseases, become dysregulated and die [9]. The ECM can function as a reservoir for growth factors and other signaling molecules, in some cases requiring MMPs to liberate them. Specifically, in the central nervous system (CNS), MMPs are evident in a variety of roles such as with brain development, synaptic plasticity, and repairment, and they have implications in different brain disorders including Alzheimer’s, multiple sclerosis, Parkinson’s disease (PD), and more [10,11]. Increased expressions of MMPs have been shown to be linked to neurodegenerative diseases such as amyotrophic lateral sclerosis (ALS), Huntington’s disease (HD), and neuroinflammatory conditions associated with PD [12,13,14]. 

A variety of models from cell-based systems, from unicellular organisms to complex organisms, have been used to study NDs [15,16,17]. The creation of animal models that recapitulate the pathology encountered in humans has been a primary driving force behind the translation of molecular insights underlying the core events in pathogenesis that lead to effective disease-modifying therapies. *Drosophila melanogaster* is an excellent model organism, with a well understood developmental biology and a relatively short life cycle, which allows for a faster experimental design process in comparison to many other model organisms [18]. The developmental genes are well-conserved in sequence and function between mammals and *Drosophila* [19]. Homologues of human diseases genes have been identified in *Drosophila*, and through biochemical analyses the functions of said genes, phenotypes, and gene products may be studied. Specifically, 77% of specific human diseases have at least one *Drosophila* homologue [20,21]. Importantly, various neurodegenerative diseases have been modeled in *Drosophila* by identifying modifier genes in disease pathology as well as employing accurate methods to assess neurodegeneration [22,23,24]. Regarding polyQ diseases, all but SCA3 and SCA7 have direct disease homologues in *Drosophila* as to humans [25]. This allows for direct comparison of molecular pathways within polyQ diseases. *Drosophila* has been shown to recapitulate the pathological features of polyQ diseases, as shown in SCA3 and HD [25,26]. In particular, the main human pathogenic features of polyQ diseases are replicated in *Drosophila*, such as: progressive neuronal degradation, cell type-specific toxicity, and age-dependent aggregates. The fruit fly, with only two MMP genes—*dMMP1* and *dMMP2*—and only one tissue inhibitor of metalloproteinase TIMP (encoded by *dTIMP*), offers an excellent model system to investigate the role of MMPs in the nervous system [27,28,29]. Importantly, in *Drosophila*, the roles of the MMPs and TIMP in age-related motor deficits, and also in neurodegenerative diseases where motor dysfunction is a common symptom, can be modeled and investigated. It has been shown previously from transcriptomic analysis that *dMMP1* expression is a bona fide biomarker of aging in motor neurons [30].

In *Drosophila*, during the process of aging, an age-dependent decline in motor function occurs, analogous to the decline experienced in mice, humans, and other mammals [31]. The molecular and cellular underpinnings of this decline are still poorly understood. It is anticipated that this age-dependent decline will be accelerated in a *Drosophila* model of SCA1 with expression of human ataxin1 with a long polyglutamine repeat of 82 amino acids in the nervous system. We hypothesized that MMP expression levels will be altered in a *Drosophila* model of SCA1 and that this in turn will impact ECM factors. Because of altered MMP–ECM interaction, there will also be changes in physiology and motor behavior. It is expected that this study will help in unraveling some of the discrete molecular mechanisms underlying the interactions of MMPs with ECM factors and their impact on physiology and behavior in neurodegenerative disease pathology.

## 2. Materials and Methods

Drosophila stocks and husbandry: All fly lines were raised on a standard cornmeal-agar medium at 25 °C and 60% ± 10% humidity under a 12:12 h light-dark cycle. All fly stocks were obtained from the Bloomington Drosophila Stock Center (Indiana University, Bloomington, IN, USA). All transgenes were out-crossed to lab reared Canton-S for at least 7 generations to isogenize them, except for the transgene constructs. The GAL4/UAS binary system [32] was used to develop a *Drosophila model* of SCA1. Stock # 33818 (P{UAS-Hsap\ATX1.82Q}M6, which expresses human Ataxin1 (ATX1 or SCA1) with a long polyQ repeat of 82 amino acids under control of UAS, was crossed to elav-GAL4 (#458 P{w[+mW.hs]=GawB}elav[C155]), a pan-neuronal driver (elav>ATX1.82Q) for driving gene expression throughout the nervous system. elav–GAL4 expression is usually restricted to post-mitotic neurons. Controls were elav-GAL4/+ and UAS-ATX1.82Q/+ where the driver or responder line was crossed to wild-type (Canton-S +/+) flies. Only male flies (F1 generation after cross) one day after eclosion were used in this study, since female flies have altered physiological status because of reproductive development, and they respond differently compared with male flies.

Longevity assay: Lifespan measurements were conducted as described before [33] in three cohorts of ca. 70 males from each cross: elav-GAL4/+; UAS-ATX1.82Q/+, and elav>ATX1.82Q. Briefly, flies were kept in polypropylene bottles (8 oz. Genessee Scientific, San Diego, CA, USA) with tissue culture dishes (60 mm Falcon Primeria from Becton Dickinson Labware, Franklin Lakes, NJ, USA) serving as lids and containers for the regular diet (15 mL). The diet dishes were replaced on alternate days after tapping the flies down to the bottom of the bottle. Mortality was scored daily, and Kaplan–Meier survival curves were generated and log-rank (Mantel–Cox) tests were conducted using GraphPad Prism 10 (v 10.2.1 www.graphpad.com, Boston, MA, USA).

Locomotor activity analysis: Flies were entrained for a day in LD 12:12 at 25 °C. Locomotor activity of 5-day-old and 30-day-old males was recorded following acclimatization for 3 days in LD 12:12, using the Trikinetics locomotor activity monitor (Trikinetics, Waltham, MA, USA) as described previously [34]. Locomotor activity was recorded for 3 days in LD 12:12, followed by 10 days in constant darkness (DD) [35]. Locomotor activity counts (actograms) were recorded by the number of infrared beam crossings of the individual flies collected in 15 min bins. The total number of beam crossings in LD cycles was averaged over a period of 3 days in LD for all flies of a genotype to generate the average daily activity profiles. Fast Fourier transform (FFT) analysis of activity data during DD was conducted using CLOCKLAB software (v 6.1.14 www.actimetrics.com, Actimetrics; Coulbourn Instruments, Whitehall Township, PA, USA) as a quantitative measure of circadian rhythmicity. Flies with FFT values < 0.04 were classified as arrhythmic, flies with FFT values ranging from 0.04 to 0.08 were categorized as weakly rhythmic, whereas flies with FFT > 0.08 were considered strongly rhythmic. Flies which showed a single peak in the periodogram (both with weak and strong rhythms) were included for calculations of free-running period using the CLOCKLAB software.

Rapid iterative negative geotaxis assay (RING): The RING assay was used to test the negative geotaxis response of males of each genotype at room temperature (25 ± 1 °C) as described previously [36,37]. Twenty-five male flies (independently) in three replicates for each age group (5 and 30 days) of each genotype were used in the RING assay. After a 3 min acclimatization period in the RING apparatus, the apparatus was rapped sharply (3X) to generate a negative geotaxis response. The climbing movements were recorded as a digital video (Appendix A). This was subsequently used for data analyses. This process was repeated five times (interspersed with ~30 s rest between trials) [37]. Trials were also randomized to ensure genotypes had sufficient recovery time. The number of flies that passed the half-way mark (5 cm) of the tubes after 5 s was analyzed and the performance was averaged and represented graphically.

Quantification of gene expression by qRT-PCR: The mRNA expression of *hATX-1* (for verification of ATX1.82Q expression), *dMMP1*, *dMMP2*, *dTIMP*, *dHh*, *dbnl*, and *dsmn* was quantified in the central nervous system (CNS–brain with ventral ganglion) of young (5-day-old) and old (30-day-old) flies in control (elav-GAL4/+ and UAS-ATX1.82Q/+) and SCA1 (elav>ATX1.82Q) flies. Total RNA was extracted from ca.100 CNS using TriReagent (Sigma-Aldrich, St. Louis, MO, USA) in three independent replicates. Samples were treated with Takara recombinant DNAse I (Clonetech labs, Mountain View, CA, USA) followed by cDNA synthesis with Iscript cDNA synthesis kit (BioRad, Hercules, CA, USA). Quantitative real-time PCR (qRT-PCR) was performed on an Applied Biosystem QuantStudio 6 Flex machine using hot-start and default thermal cycling conditions with data acquisition at the end of the extension step, followed by a dissociation step and melt curve analysis. Every reaction contained Power SYBR Green (Applied Biosystem, Carlsbad, CA, USA), 10 ng of cDNA and 400 nM of primers. The primer sequences used have been listed in Appendix A. The primers were checked for their efficiency using 10^(−1/slope)^ − 1 and the slopes obtained were between −3.46 and −3.39 (94–97% efficiency). Data were analyzed using the 2^−ΔΔCT^ method [38] with mRNA levels normalized to the housekeeping gene *rp49*. Since there was no significant difference in gene expression in control flies (elav-GAL4/+ and UAS-ATX1.82Q), the expression data were averaged for a specific age group. Relative expression was calculated with respect to the averaged control day 5 data set as 1.

Western blotting: Three independent bioreplicates of 30-day-old males of controls (elav-GAL4/+ and UAS-ATX1.82Q/+) and SCA1 (elav>ATX1.82Q) were collected. CNS (Brain and ventral ganglion) tissue was dissected out from 50 flies that were homogenized on ice in 50 mM phosphate buffer, sonicated, and centrifuged at 10,000× *g* for 10 min at 4 °C. The protein content was equalized to ensure equal protein loading using the bicinchoninic acid method [39]. Samples were reconstituted with 4X Laemmli sample buffer and boiled for 10 min and then separated by polyacrylamide gel electrophoresis (SDS-PAGE) on 7.5% resolving gel [40], followed by transfer onto PVDF Immobilon membranes (MilliporeSigma, Burlington, MA, USA) and incubation in 1 × TBST (10 mM Tris, 0.15 M NaCl, 0.1% Tween-20, pH 7.5) + 5% milk for 2 h. The membranes were then incubated overnight at 4 °C with primary antibody 1:100 Mouse anti-dMMP1 (Developmental Studies Hybridoma Bank) and 1:500 for dMMP2 and dTIMP, which were both generated in rabbits (custom-made by Abcam, MA, USA ) in blocking buffer. Alpha-tubulin was employed as loading control and the antibody (1:1000) was developed in rabbits (Cell Signaling Technology, Danvers, MA, USA). Membranes were treated for 2 h with 1:20,000 with Donkey anti-mouse IRDye 680RD or goat anti-rabbit IRDye 680RD (LI-COR Biosciences, Lincoln, NE, USA). Blots were scanned using the LI-COR Odyssey Infrared Imaging System (CLx, LI-COR Biosciences, Lincoln, NE, USA ) and quantified with imaging software (Image Studio, v. 3.0, LI-COR Biosciences, Lincoln, NE, USA).

Statistical analysis: Statistical analysis of longevity was conducted using the log-rank (Mantel–Cox) and Gehan–Breslow–Wilcoxon tests to analyze the Kaplan–Meier survival curves. For daily locomotor activity graphs and Western blot analysis relative quantitation, a one-way ANOVA with Brown–Forsythe test and Barlett’s test (with Dunnett’s multiple comparison test) and an unpaired t-test were used. For RING and gene expression analysis, a two-way ANOVA (with Bonferroni post-test) was employed. Unless otherwise indicated, statistical significance was set at *p* < 0.05. All statistical analysis and graphs were generated using GraphPad Prism 10 v 10.2.2 (GraphPad Software, Inc., Boston, MA, USA).

## 3. Results

### 3.1. Pan-Neuronal Expression of Human ATX1.82Q Shortens Lifespan

Longevity assays were conducted to document the impact of expression of human ATX1.82Q in flies. Pan-neuronal expression of human ATX1.82Q resulted in a significant (*p* < 0.0001) shortening of lifespan (48 ± 2 days) compared to control flies (elav-GAL4/+: 67 ± 2 days and UAS-ATX1.82Q/+: 69 ± 1.5 days) (Figure 1).

### 3.2. Pan-Neuronal Expression of Human ATX1.82Q Does Not Impact Diurnal Locomotor Rhythms in Young Flies but Dampens Diurnal Anticipatory Behavior in Old Flies

Daily diurnal locomotor activity rhythms were recorded at day 5 and day 30 and plotted to compare the activity patterns of flies expressing human ATX1.82Q with those of control flies (eval-GAL4/+ and UAS-ATX1.82Q/+). Interestingly, while young (5-day-old) flies did not show any difference in the diurnal locomotor activity profiles of controls (elav-GAL4/+, UAS-ATX1/+) and SCA1 (elav>ATX1.82Q) (Figure 2A,C,E), old (30-day-old) flies showed a marked dampening of anticipatory enhancement of activity in response to night-day transition and day-night transition in SCA1 flies (Figure 2F) compared to controls (Figure 2B,D).

### 3.3. Expression of Human ATX1.82Q Results in Decreased Horizontal Daily Diurnal Locomotor Activity in Old Age

The average daily activity of control (elav-GAL4/+ and UAS-ATX1.82Q) and SCA1 (eval>ATX1.82Q) flies was plotted in LD cycles for 5-day-old and 30-day-old flies (Figure 3A,B). While no significant difference in activity was observed in young (5-day-old) flies (Figure 3A), in old (30-day-old) SCA1 flies (elav>ATX1.82Q) the activity was significantly (*p* < 0.0001) less compared to controls (elav-GAL4/+ and UAS-ATX1.82Q/+) (Figure 3B). A reduction in activity of 32 ± 2.8% (mean ± SD) was recorded among the controls from day 5 to day 30, whereas in case of SCA1 flies this decrease in activity was 44 ± 5.7%. Furthermore, 30-day-old flies SCA1 flies showed 26 ± 3.2% less activity compared to controls.

### 3.4. Negative Geotaxis Is Significantly Impaired in Young and Old Flies Expressing Human ATX1.82

Negative geotaxis was assessed in young (5-day-old) and old (30-day-old) flies. SCA1 flies (elav>ATX1.82Q) showed significantly (*p* < 0.05) reduced negative geotaxis response compared to controls (elav-GAL4/+ and UAS-ATX1.82Q) in both age groups (Figure 4). Importantly, a general decline in negative geotaxis was observed between young and old flies of both controls and SCA1 flies (Figure 4).

### 3.5. Flies Expressing Human ATX1.82Q Exhibit a Significant Upregulation in Expression of dMMP1 in Young and Old Flies and a Significant Decline in Expression of Extracellular Matrix Fibroblast Growth Factors and Survival Motor Neuron Gene in Old Flies

Since motor function in SCA1 (elav>ATX1.82Q) was impaired, as exhibited by a decline in average daily activity and negative geotaxis, we decided to check if matrix metalloproteinases (implicated in motor deficits with age), ECM factors, and survival motor neuron (SMN) protein could be affected. Indeed, the expression of *dMMP1* was significantly elevated in an age-dependent manner in both control and SCA1 flies (Figure 5A). The increase in *dMMP1* expression was greater in SCA1 flies compared to controls in both age groups. Interestingly, *dMMP2* and *dTIMP* were marginally increased in 30-da- old SCA1 flies (Figure 5B,C), while no changes were recorded in young flies of all genotypes or young and old flies of controls.

ECM factors and receptors have been implicated in neurogenesis, neuroplasticity, and repair [41]. We decided to check if there is any change in the expression of some key ECM factors. The ECM fibroblast growth factor (FGF) hedgehog (*dHg*) and branchless (*dbnl*) expression showed a marked decline with age in both controls and SCA1 flies (Figure 5D,E). Importantly, the gene *dsmn* encoding for the survival motor neuron protein [42] was significantly expressed less in old SCA1 flies compared to controls which also showed a decline with age (Figure 5F).

### 3.6. dMMP1 Protein Levels Were Markedly Increased in 30-Day-Old Flies Expressing Human ATX1.82Q Compared to Controls but Not dMMP2 and TIMP

To check if increased transcription of *dMMP1*, *dMMP2* and *dTIMP* in old SCA1 flies is reflected in protein levels, we conducted Western blots for the proteins in 30-day-old control (pooled elav-GAL4/+ and UAS-ATX1.82Q/+) and SCA1 (elav>ATX1.82Q) flies. Interestingly, only dMMP1 protein levels were significantly (*p* < 0.0001) elevated in SCA1 flies compared to controls (Figure 6A,B, Appendix A), whereas no change in protein levels of dMMP2 or dTIMP in 30-day-old SCA1 vs control flies was observed) (Figure 6A,B).

## 4. Discussion

Spinocerebellar ataxia type 1(SCA1) is a progressive neurodegenerative disease which is characterized by loss of motor coordination (ataxia) and balance. This is caused by an expansion in the number of CAG (encoding for glutamine) repeats in the coding region of *Atx-1* (or *ATXN-1*) gene [6]. In this study, a *Drosophila* model of SCA1 was developed by driving the expression of human ATX1 with a long polyQ (glutamine) repeat of 82 amino acids under the control of UAS using a pan-neuronal driver elav-GAL4. We observed a significant reduction of lifespan in flies expressing the human ATX1.82Q compared to controls (where the driver or responder line was crossed to wild type) (Figure 1). Interestingly, this contradicts an earlier observation that elav-GAL4-driven expression of ATX1.82Q does not impact lifespan [43]. There could be two possibilities on why we observed a different effect: (a) the earlier study was conducted on pooled male and female flies, whereas the present study focused on only male flies; and (b) the elav-GAL4 driver line we used in this study is C155, the one most used and originally described, which is linked to the X chromosome. In fact, in their study, the authors mentioned clearly that their elav-Gal4 is not C155, but a minor type linked to the second chromosome (P{GAL4-elav.L}2) which did not show homogenous pan-neuronal expression [43]. We verified the expression of *hATX-1* (encoding for ATX-1.82Q) in CNS by gene expression (Appendix A), which was present only in SCA1 flies and absent in age-matched controls. To examine if pan-neuronal expression of ATX1.82Q results in deficits in diurnal locomotor activity rhythms, we checked the locomotor activity patterns in the light:dark (LD) cycle in both young and old flies. In young flies (5-day-old), we could not find any difference in anticipatory peak in activities during the transition from night to day or day to night in SCA1 flies (elav>ATX1.82Q) compared to controls (elav-GAL4/+ and UAS-ATX1.82/+) (Figure 2A,C,E), which agreed with the results reported earlier [43]. However, in 30-day-old flies, this anticipatory peak in activity was significantly dampened in SCA1 flies (Figure 2F) compared to controls (Figure 2B,D). Thus, we did see an age-dependent effect of pan-neuronal expression of ATX1.82Q on diurnal locomotor activity rhythms, which was not observed in the earlier study. Additionally, we also observed that while young (5-day-old) control flies were 100% rhythmic in total darkness (DD), the SCA1 flies were only 67% rhythmic (Appendix A). The rhythmicity declined with age (30-day-old) in controls (56% rhythmic), whereas in SCA1 flies it was even less (23%) (Appendix A). This indicates an age-dependent effect on circadian locomotory patterns in SCA1 flies. We examined the circadian expression of core clock genes in the heads of 30-day-old SCA1 flies compared to age-matched controls given the significant decline in rhythmicity. We observed a significant dampening in the amplitude of expression of core clock genes *period* and *timeless* in SCA1 compared to controls, but *Clock* and *Cycle* expression remained unchanged (Appendix A).

Aging involves a gradual transition from youthful vigor to geriatric infirmity and death [44]. Age-associated behavioral senescence can be effectively measured in adult *Drosophila* [31]. Much like other organisms, the fly also exhibits an age-associated decline in locomotor activity [30,45,46,47]. Many behaviors including locomotor behavior, learning, and memory are known to decline during aging in *Drosophila* [31,48,49]. However, in neurodegeneration-prone mutants, the decline in such behaviors is accelerated. In our present study, the total daily activity was not significantly different in young (5-day-old) SCA1 flies compared to controls (Figure 3A). This was also in agreement with results reported earlier [43]. However, in 30-day-old flies, the total daily activity of SCA1 flies was significantly less than that of the controls (Figure 3B). This significant decline in total daily activity in old SCA1 flies can be attributed to possible effects on motor neurons [50].

To examine if this effect on motor neurons extended to another robust test of functional decline, we performed the widely employed rapid iterative negative geotaxis (RING) assay [36]. In this case, we did see a significant decline in negative geotaxis response of SCA1 flies both in young (5-day-old) and old (30-day-old) SCA1 flies compared to controls (Figure 4). These results again differed from those reported earlier by Shairishi et al. [43], who examined motor performance in 7-day-old and 14-day-old SCA1 flies. Interestingly, they also did not find any difference in motor performance of control elav-GAL4/+ flies in the 7- and 14-day time span. Importantly, they did not check if such decline occurs in older flies (30 days old) as we did in this study.

This rapid and significant functional decline in motor performance in SCA1 flies led us to question whether motor neuron function is impaired especially at the neuromuscular junction (NMJ) by matrix metalloproteinase as demonstrated earlier [30]. Matrix metalloproteinases (MMPs) perform several critical functions in the CNS ranging from growth and development to neural plasticity, repair as well as in neurodegenerative diseases [29,51,52]. Age-dependent motor decline with increased expression of matrix metalloproteinase 1 (*dMMP1*) gene as reported earlier by Azpurua et al. [30] led us to inquire if MMPs expression could be a likely contributing factor for the age- and neurodegeneration-associated decline in motor responses. Indeed, we found that *dMMP1* expression was significantly upregulated in an age-dependent manner in both controls (elav-GAL4/+ and UAS-ATX1.82Q) and in SCA1 (elav>ATX1.82Q) flies. Also, in each age group, the SCA1 flies showed significantly higher expression of *dMMP1* compared to controls (Figure 5A). Interestingly, while *dMMP1* expression levels were marginally elevated in 5-day-old SCA1 flies compared to controls, this did not translate to elevated protein levels. On the other hand, *dMMP2* expression was marginally and significantly upregulated in old SCA1 flies compared to controls (Figure 5B). However, when protein levels were examined using Western blotting, only dMMP1 protein levels were significantly elevated compared to controls in 30-day-old flies (Figure 6A,B), whereas dMMP2 protein levels remained unchanged. A plausible explanation for no change in dMMP2 protein could be post-transcriptional changes/modifications that result in unchanged protein levels despite higher transcriptional expression of *dMMP2* in 30-day-old SCA1 flies. Interestingly, the tissue inhibitor of matrix metalloproteinase (dTIMP) also mirrored the dMMP2 in both protein and expression levels (Figure 5C and Figure 6A,B). TIMP can inhibit both dMMP1 and dMMP2, however, because dTIMP levels are unchanged, the dMMP1 levels are elevated, and it is also possible that dMMP2 levels are regulated to some extent by dTIMP levels ensuring a status quo. In general, MMP expression levels are highly elevated in a number of neuronal pathologies. MMP upregulation in CNS disease states raises the issue of whether MMP induction has an overall positive or negative effect on disease outcome. There is substantial evidence that the net effect of high MMP expression in some diseases is detrimental [53,54].

In flies, dMMP1 is required throughout the peripheral nervous system for proper development including axon fasciculation and remodeling of both dendrites and nerve terminals [30,55]. Following development, dMMP1 expression is downregulated in the motor neurons, but this repression is lost with age and in neurodegenerative diseases, with deleterious consequences in motor responses. The precise mechanism by which dMMP1 causes motor neuron dysfunction is unclear, but we did record an elevation in expression of the *dbsk* (*basket*) gene, which encodes for c-Jun N-terminal kinase (JNK), which in turn plays a role in neuronal connectivity [56] and could result in elevated dMMP1 (Appendix A).

MMPs are able to cleave nearly every component of the extracellular matrix (ECM), as well as numerous signaling molecules and cell surface receptors [57]. Fibroblast growth factors (FGF) are essential signaling proteins that regulate diverse cellular functions in developmental and metabolic processes. To investigate if ECM factors, i.e., FGF ligands such as Hedgehog (encoded by *dHh*) and Branchless (*dbnl*), could be impacted by dMMP1 at the NMJ, we checked their expression levels in the CNS (brain and ventral ganglion). Both *dHh* and *dbnl* showed a decline in expression in 30-day-old flies when compared to young (5-day-old) files (Figure 5D,E). Moreover, 30-day-old SCA1 flies showed a significant decline in both *dHh* and *dbnl* expression compared to age-matched controls. It has been demonstrated earlier that the depletion of Hedgehog reduces lifespan and impacts locomotor activity [58]. Also, the expanded polyQ tract in SCA1 flies triggers a pathogenic cascade, leading to selective neuronal cell dysfunction and death [59,60]. It has also been reported that polyQ ataxin3 alters specific gene expression through DNA methylation and one of the targets is Hedgehog signaling in SCA3/MJD mice models [61]. In *Drosophila*, the FGF homolog, *branchless* (*bnl*) is expressed in a dynamic and spatiotemporally restricted pattern to induce branching morphogenesis of the trachea, which expresses the Bnl-receptor, *breathless* (*btl*). Genetic evidence suggests that Bnl-signaling is also important for several non-tracheal morphogenesis events. These functions include control of cell adhesion in the embryonic midline epithelia and in the developing retina, cell proliferation in larval neuroblasts, and migration of glial cells in the embryonic central nervous system (CNS) [62,63]. We speculate that there is a possible role for Hh and bnl signaling in normal motor neuron health and polyQ toxicity and that aging impairs FGF receptor signaling. 

In vertebrates, synaptic development and maintenance use at least three distinct signaling mechanisms: the TGF β; *wingless*; and FGF pathways. In *Drosophila*, it is noteworthy that the first two have been demonstrated to function in a similar fashion at the NMJ [64,65]. The survival motor neuron (SMN) is an essential protein that is classed as ubiquitous and functions in the biogenesis of an assembly chaperone for Sm-class small nuclear ribonucleoproteins in the SMN complex which mediate pre-mRNA splicing [66]. The SMN complex contains three other 'Gemin' proteins encoded by Gem2, Gem3, and rig. Smn is essential for viability, but higher levels of the protein are required for proper neuromuscular development and function. SMN depletion early during neurogenesis has been implicated with motor defects in a *Drosophila* model of spinal muscular atrophy (SMA) [67]. While *dsmn* expression has not been demonstrated in adult *Drosophila*, expression-level data in adult stages demonstrate high expression levels in males and females (modENCODE RNA-seq–Flybase http://flybase.org/reports/FBgn0036641, accessed on 29 March 2024). Also, Fly Cell Atlas scRNA-seq data show low to moderate expression levels in adult motor neurons and neurons in general. As such, we decided to investigate whether SMN depletion could occur during aging and whether it is also associated with a polyQ disease. Interestingly, *dsmn* expression mirrored the FGF factors (*dHh* and *dbnl*) expression. It has also been demonstrated earlier that SMN activity regulates the expression of FGF signaling components and thus FGF signaling [58]. Furthermore, they showed that alterations in FGF signaling activity are able to modify the NMJ defects caused by loss of *dsmn* function [68]. While these studies were conducted in larval stages, we hypothesize that, in adult stages with aging, there is dysregulation in SMN activity and FGF signaling in general, and that this gets compounded by neurodegenerative diseases such as SCA1, leading to motor deficits. Our working hypothesis based on the results of this investigation is that, while MMP1 levels *per se* do not contribute to disease initiation in SCA1, it could rather be that polyQ toxicity contributes to the dysregulation of MMP1, which then impairs motor ability.

## 5. Conclusions

Taken together, our data suggest that aging and polyQ toxicity could increase MMP1 levels through the inflammatory JNK pathway (*dbsk*), which either directly or indirectly results in decreased expression of *dsmn* and FGF factors such as *dHh* and *dbnl*. The net result of such dysregulation is accelerated motor deficits in polyQ disease (Figure 7). Further research will be required to unravel the precise molecular pathways by which polyQ disease such as SCA1 leads to accelerated motor deficits.

## Figures and Tables

**Figure 1 biology-13-00854-f001:**
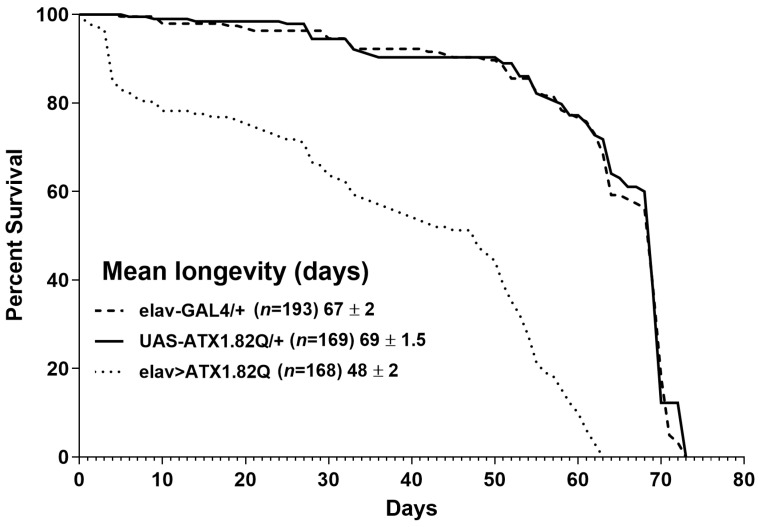
Longevity of flies. Kaplan–Meier survival curves of elav-GAL4/+, UAS-ATX1.82Q/+ and elav>ATX1.82Q male flies under 12 h light:dark (LD) cycles and *ad libitum* feeding conditions. The log-rank (Mantel–Cox) test and the Gehan–Breslow–Wilcoxon test revealed significant differences (*p* < 0.0001) between survival curves of elav>ATX1.82Q and controls (elav-GAL4/+ and UAS-ATX1.82Q/+. No difference in survival curves between elav-GAL4/+ and UAS-ATX1.82Q/+ flies was recorded. Survival data were obtained from three independent replicates of approximately 70 flies of each genotype and have been depicted as mean longevity (days) ± SD.

**Figure 2 biology-13-00854-f002:**
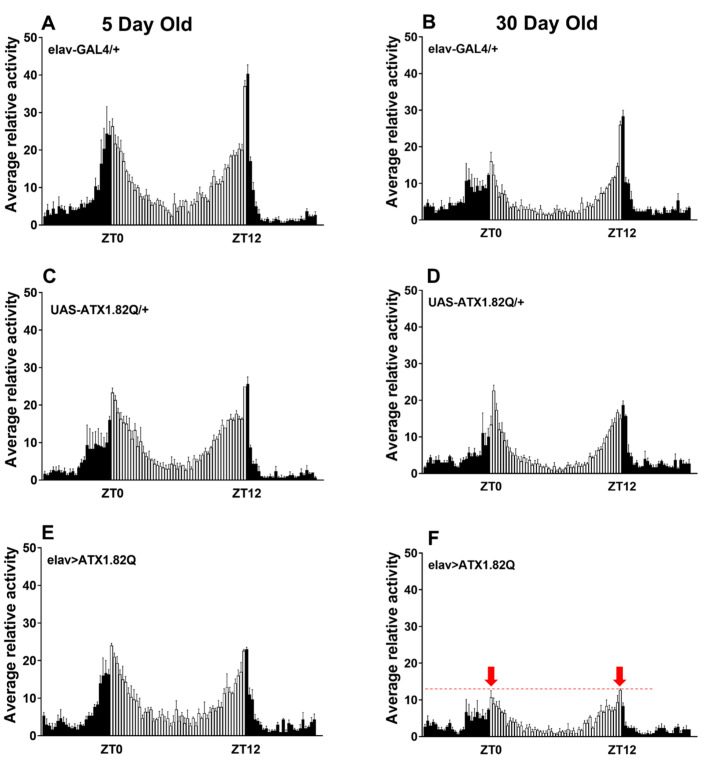
Diurnal locomotor activity pattern. Daily locomotor activity patterns in LD cycles were recorded as 15 min bin beam breaks over a period of 3 days, averaged, and plotted over a 24 h period. A minimum of 32 flies and a maximum of 64 flies of each genotype were tested. The locomotor activity in LD cycles was recorded for all three fly genotypes at young (5-day) and old (30-day) ages: (**A**) elav-GAL4/+ 5-day-old; (**B**) elav-GAL4/+ 30-day-old; (**C**) UAS-ATX1/+ 5-day-old; (**D**) UAS-ATX1.82Q/+ 30-day-old; (**E**) elav>ATX1.82Q 5-day-old; and (**F**) elav>ATX1.82Q 30-day-old. Dark bars represent scotophase (night) while white bars represent photophase (day). Zeitgeber Time ZT 0 is 9 am and ZT 12 is 9 pm. The red dotted line on (**F**) denotes the level of attenuation of the anticipatory peak (marked by red arrows) in 30-day-old elav>ATX1.82Q flies. Data are represented as mean ± SD.

**Figure 3 biology-13-00854-f003:**
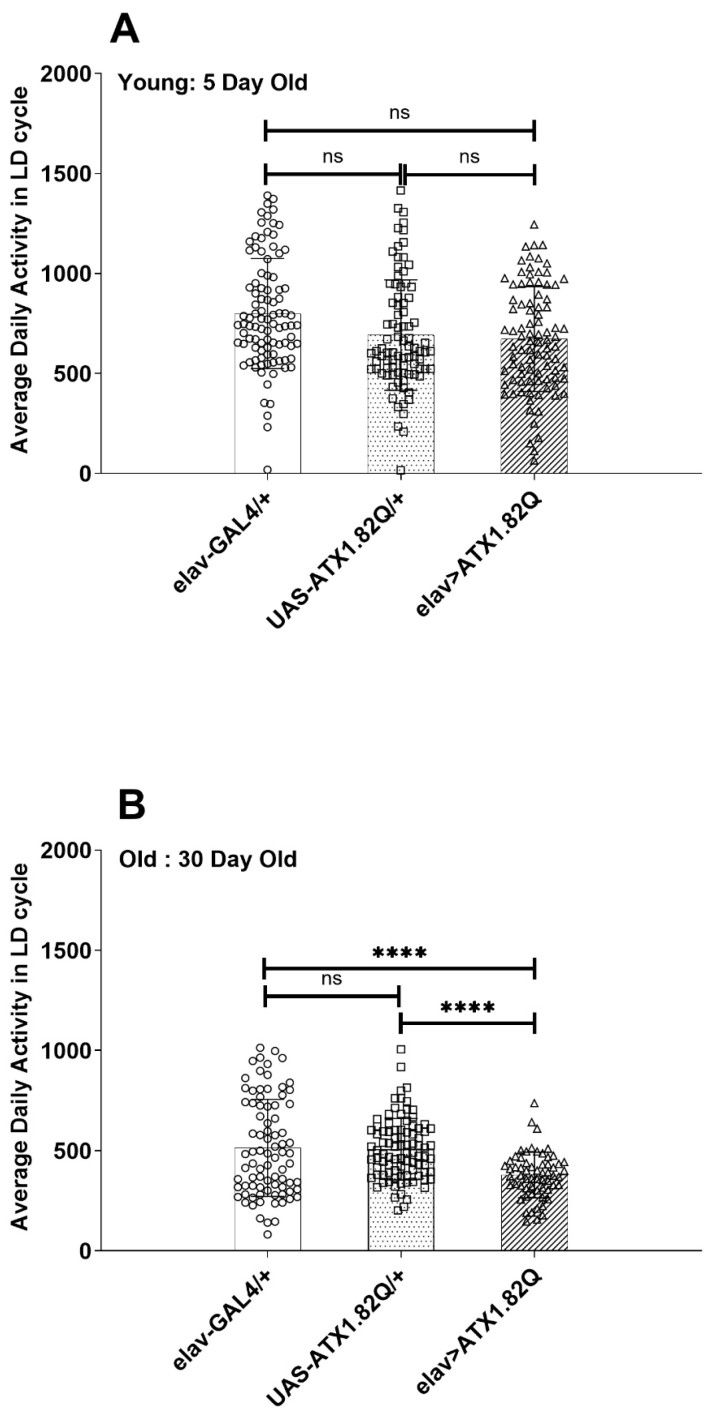
Diurnal activity levels. Average daily activity of flies of each genotype in LD cycles was plotted for (**A**) 5-day-old (**B**) 30-day-old flies. The total beam breaks for each fly of each genotype were summed up and averaged by the total number of flies tested, and average daily activity in terms of beam crossings was calculated. One-way ANOVA with Brown–Fosythe test (*p*-value = 0.9656) and Barlett’s test (*p*-value = 0.8640) with Dunnett’s multiple comparison test revealed no significant difference in 5-day-old flies’ average daily activity (**A**). In-30-day old flies (**B**), however, a significant difference was recorded between the controls’ (elav-GAL4/+ and UAS-ATX1.82Q/+) and elav>ATX1.82Q flies’ average daily activity (one-way ANOVA with Brown–Forsythe and Bartlett’s test with Dunnett’s multiple comparison test, **** = *p* < 0.0001). Data have been plotted showing the spread of individual values (mean ± SD). Blank circles represent elav-GAL4/+; blank squares represent UAS-ATX1.82Q/+ and blank triangles represent elav>ATX1.82Q.

**Figure 4 biology-13-00854-f004:**
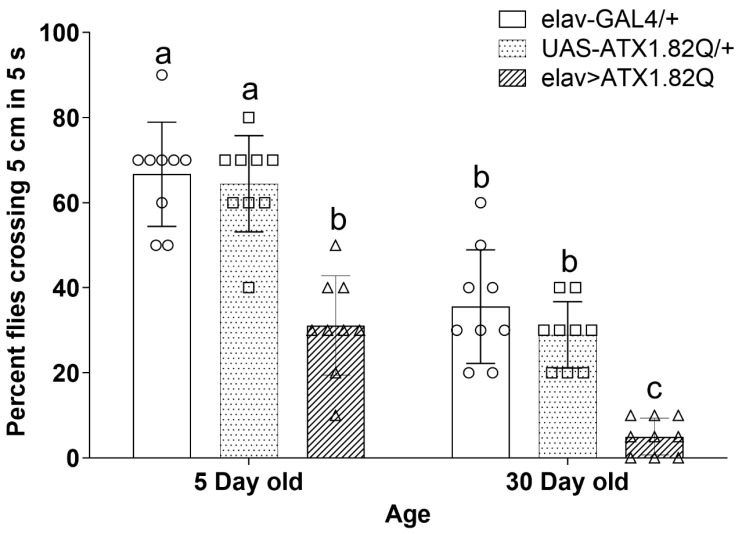
Rapid iterative negative geotaxis. Negative geotaxis (RING) assay of experimental (young ~ 5-day-old; old ~ 30-day-old) male flies of various genotypes. Bar graphs represent the percentage of flies crossing t 5 cm mark in a period of 5 seconds. Data are presented showing the spread of the individual mean values of three independent repeats ± SD (*n* = 75 flies of each genotype and age). Data were analyzed by two-way ANOVA with genotype and age being fixed effects, followed by a Bonferroni post hoc test. Comparisons were made among genotypes within a specific age and also among genotypes across different ages. Bars with different superscripts (small alphabets) are significantly different at *p* < 0.05. Blank circles represent elav-GAL4/+; blank squares represent UAS-ATX1.82Q/+ and blank triangles represent elav>ATX1.82Q.

**Figure 5 biology-13-00854-f005:**
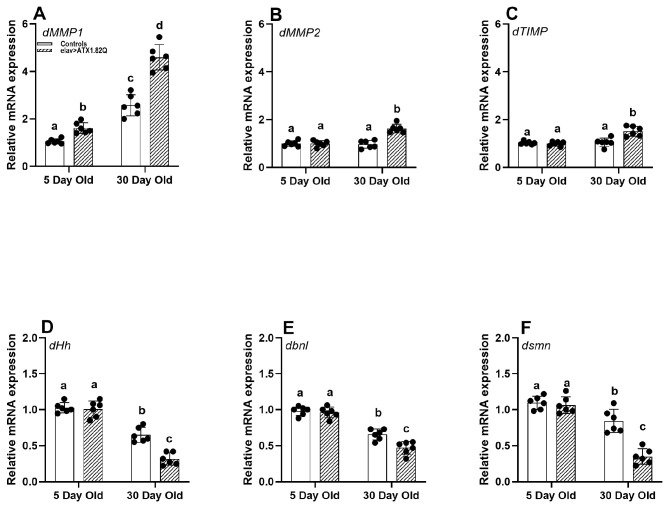
Gene expression analysis. Relative gene (qPCR) expression of (**A**) *dMMP1*, (**B**) *dMMP2*, (**C**) *dTIMP*, (**D**) *dHh*, (**E**) *dbnl*, and (**F**) *dsmn* genes in males of young (5-day-old) and old (30-day-old) flies of controls (pooled elav-GAL4/+, UAS-ATX1.82Q/+) and elav>ATX1.92Q flies. mRNA expression of control (5-day-old) was set as a reference (=1). Values represent mean ± SEM of three independent bioreplicates with three technical replicates with the spread of mean values (*n* = 100 CNS from flies of each genotype and age). Two-way ANOVA with the Bonferroni post hoc multiple comparisons test was conducted to separate out the means. Comparisons were made between genotypes (within an age) as well as across age (between genotypes). Bars with different superscripts (small alphabets) are significantly different at *p* < 0.05.

**Figure 6 biology-13-00854-f006:**
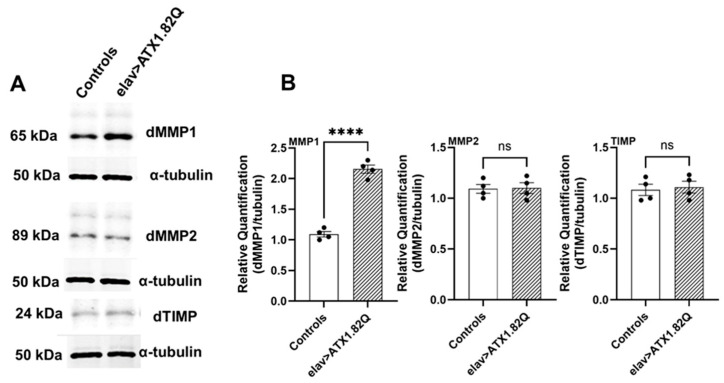
Western blots of dMMP1, dMMP2 and dTIMP. Western blots were performed for (**A**) dMMP1, dMMP2, and dTIMP in 30-day-old control (pooled elav-GAL4/+ and UAS-ATX1.82Q/+) and elav>ATX1.82Q flies. Alpha-tubulin was used as the loading control. Figures are representative of one of three independent blots. (**B**) Blots were quantified by mean gray scale value relative to tubulin, and bars with individual data spread are represented as mean ± SD of three independent blots. An unpaired *t*-test was conducted to test for differences among the means. (**** = *p* < 0.0001).

**Figure 7 biology-13-00854-f007:**
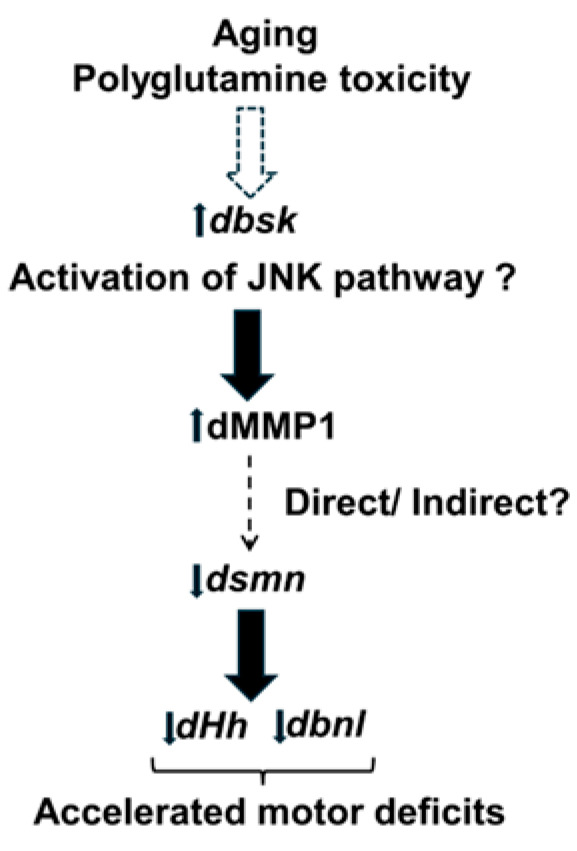
Hypothetical pathway leading to accelerated motor deficits. Aging and Polyglutamine toxicity could lead to activation of the c-Jun kinase (JNK) pathway by enhanced expression of *dbsk* (basket). This would lead to the observed increase in dMMP1 levels, which could directly or indirectly downregulate *dsmn* (survival motor neuron) expression, which in turn would downregulate ECM FGF factors such as *dHh* (hedgehog) and *dbnl* (branchless) at the neuromuscular junction or ventral ganglion, leading to accelerated motor deficits. Dotted arrows represent pathways that could be direct or indirect, whereas solid arrows represent a direct regulatory effect.

## Data Availability

The original contributions presented in the study are included in the article/Appendix A. Further inquiries can be directed to the corresponding author.

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
