# Peer review of "Enhanced Age-Dependent Motor Impairment in Males of Drosophila melanogaster Modeling Spinocerebellar Ataxia Type 1 Is Linked to Dysregulation of a Matrix Metalloproteinase"

_biology, 2024, doi:10.3390/biology13110854_

Round 1
Reviewer 1 Report
Comments and Suggestions for Authors
Emma M. Palmer et al. demonstrated that expressing human Ataxin-1 with a long polyQ repeat in Drosophila neurons resulted in a shortened lifespan and impaired motor function in male flies. These effects were linked to elevated levels of matrix metalloproteinase 1 (dMMP1), diminished extracellular matrix signaling, and reduced expression of the survival motor neuron gene, suggesting that matrix metalloproteinase dysregulation may contribute to polyQ diseases. It is a well-conducted study, and I have two major points:
- Could you provide more evidence showing how changes in matrix metalloproteinase levels contribute to disease initiation? For instance, what is the expression pattern of dMMP1 in middle-aged Drosophila?
- Have you performed Western blots to validate the protein levels of dMMP1 in young flies, given that its mRNA expression is upregulated at that stage?
Author Response
We thank the reviewer for the valuable and insightful comments on the manuscript and outline our responses below to the concerns raised by the reviewer as well as our responses and action taken in the revised manuscript.
Comment 1: Could you provide more evidence showing how changes in matrix metalloproteinase levels contribute to disease initiation? For instance, what is the expression pattern of dMMP1 in middle-aged Drosophila?
Response: We would like to clarify that dysregulation of matrix metalloproteinase contributes to one of the overt symptoms associated with SCA1 which is ataxia or motor impairment. However, the disease initiation process by polyglutamine toxicity has not been the focus of this investigation, rather how locomotor ability is impaired as a result of the disease by dysregulation of MMP1. To our knowledge, MMP1 levels per se do not contribute to disease process, rather, MMP1 levels are affected by the disease. We have now changed this in the Simple Summary (Line 17-18) and this has been emphasized as well in the Discussion section (474-477) of the revised manuscript. The expression pattern of MMP1 in middle aged (15 day old) Drosophila was elevated as was observed in young flies, but not to the extent that was observed in 30 day old flies. We specifically chose the 30 day fly group when locomotor activity patterns were significantly dampened in SCA1 as a read-out of SCA1’s overt effects on motor activity.
Comment 2: Have you performed Western blots to validate the protein levels of dMMP1 in young flies, given that its mRNA expression is upregulated at that stage?
Response: That is an excellent point! We did check if dMMP1 protein levels are increased in 5 day old flies, since the expression level was marginally increased in SCA1 flies compared to controls. However, we did not get a consistent up-regulation of protein levels in the Western blots that we performed. We assume that there may be post-transcriptional factors that could have regulated its levels along with TIMP, which inhibits dMMP1. We have now mentioned this in the Discussion (Lines 403-404) of the revised manuscript.
Reviewer 2 Report
Comments and Suggestions for Authors
This manuscript used fruit flies to model Spinocerebellar Ataxia Type 1, showing that expressing human Ataxin-1 with long polyQ repeats shortens lifespan and worsens motor skills with age due to changes in matrix metalloproteinase and extracellular matrix signaling. Some suggestions are added below.
1. The authors used genetically modified fruit flies to model Spinocerebellar ataxia Type 1. However, it would be helpful to explain why they did not check the expression of human Ataxin-1 (ATX1.82Q) in the brain or consider using the eye, as some studies have done, to demonstrate the toxicity of this protein.
2. The authors have not provided references for many methods used in the experimental design section. It would be helpful to clarify where these methods were taken from previous studies or developed original by author.
3. The manuscript suggests that toxicity in ataxia type 1 increases with changes in matrix metalloproteinase with age. Since this mechanism has been explored in many studies, could the authors clarify the novelty of their findings?
4. If the authors propose a specific mechanism, why did they not use mutant forms of these proteins to confirm and strengthen the conclusions of the manuscript?
5. Including the experiment related to apoptosis protein expression change measurement like caspases (as the alterations in suggested proteins and genes are associated with cell death), could enhance reader interest and provide deeper insights into the implications of the findings.
6. In negative Geotoxic assay the author gave 30s interval for each trial in groups, please add the related references.
7. Adding videos that demonstrate locomotory changes could further strengthen the manuscript and provide a clearer visual representation of the findings.
8. Author selected the two-time period 5day and 30day for the almost experiment, why author did not check the these in 15 days, please explain.
Author Response
Reviewer 2:
We thank the reviewer for the insightful and useful comments that has helped us to improve the manuscript. We outline below our responses/ clarifications to comments as well as action taken in the revised manuscript:
Comment 1. The authors used genetically modified fruit flies to model Spinocerebellar ataxia Type 1. However, it would be helpful to explain why they did not check the expression of human Ataxin-1 (ATX1.82Q) in the brain or consider using the eye, as some studies have done, to demonstrate the toxicity of this protein.
Response: We did check ATX-1 expression levels in the fly brain and as was expected it was significantly elevated. We are currently working on a manuscript that addresses the aspect of ATX-1 localization and PolyQ toxicity. This aspect of part of that manuscript which also deals with neurodegeneration. In this study our aim was to understand the primary cause of ataxia associated with SCA1, and our focus was on behavioral characterization coupled with a plausible mechanism. We have also employed the eye-specific GMR-Gal4 driver in that study and have documented polyQ toxicity in the eye.
Comment 2. The authors have not provided references for many methods used in the experimental design section. It would be helpful to clarify where these methods were taken from previous studies or developed original by author.
Response: We have provided references in the methods for e.g. for Longevity assays: Reference 33: Bednarova et al., 2018; Locomotor activity analysis: Reference 34: Bednarova et al., 2017; RING assay: References 36 and 37. The method was originally developed by Gargano et al., 2005 [36] and modified by the author(s) Krishnan et al., 2012 [37]. For qPCR and Western blotting, the cited references are 38, 39 and 40: Livak and Schmittgen, 2001; Smith et al., 1995 and Laemmli, 1970.
Comment 3. The manuscript suggests that toxicity in ataxia type 1 increases with changes in matrix metalloproteinase with age. Since this mechanism has been explored in many studies, could the authors clarify the novelty of their findings?
Response: An excellent point! As far as we are aware, our study is the first that links dysregulation of MMP1 to impairment in motor activity associated with a polyQ disease.
Comment 4. If the authors propose a specific mechanism, why did they not use mutant forms of these proteins to confirm and strengthen the conclusions of the manuscript?
Response: We thank the reviewer for this important suggestion. We are currently working on genetic recombination studies where in transgenic flies expression ATX-1.82Q we can also have a conditional knockdown of MMP1 and check if that rescues the motor response. In this study, our main focus was to lay out how dysregulation of MMP1 is associated with motor impairment in an SCA1 disease model, the next logical step is to leverage the power of Drosophila genetics to manipulate the target genes/proteins involved in the process.
Comment 5. Including the experiment related to apoptosis protein expression change measurement like caspases (as the alterations in suggested proteins and genes are associated with cell death), could enhance reader interest and provide deeper insights into the implications of the findings.
Response: We appreciate this useful suggestion and that is precisely the focus of our next publication where we focus on Dronc, DREDD (caspase) and Eiger as well as SUMOylation process in polyQ toxicity.
Comment 6. In negative Geotoxic assay the author gave 30s interval for each trial in groups, please add the related references.
Response: The original method for the RING assay has been cited [36], together with modifications done by the author(s) in 2012 [37] which is what has been done in this study.
Comment 7. Adding videos that demonstrate locomotory changes could further strengthen the manuscript and provide a clearer visual representation of the findings.
Response: We have the RING assay videos, but the locomotor activity assays are conducted in L:D cycle followed by 7 days of complete darkness where locomotor activity would be hard to monitor except as Trikinetics readout.
Comment 8. Author selected the two-time period 5day and 30day for the almost experiment, why author did not check the these in 15 days, please explain.
Response: An excellent question. We chose young (5-day old) and old (3-day old) flies, rather than middle aged flies (15 days) because locomotor activity dampening was significantly impacted only in the 30 day old age cohort. The 15 day old SCA1 flies had rhythmic locomotor activity as 5 day old SCA1 flies and activity levels were not much different from age-matched controls.
Round 2
Reviewer 2 Report
Comments and Suggestions for Authors
This manuscript exploring the role MMP1 mechanism in poly q toxicity by using drosophila model. Some minor suggestions are added below.
11. Adding the results of ATX1 expression in the manuscript could strengthen and increase the reliability of the findings.
2. Many studies suggest that locomotor alterations begin around 15 days, but in the context of the mechanism, selecting 30 days is also considerable.
3. Including the ring assay video might enhance the strength of the manuscript.
4. Overall, the manuscript is well-designed and provides mechanistic insights into Poly-Q diseases.
Author Response
We thank the reviewer for suggesting additional edits/ revisions to the manuscript and we have accordingly revised the manuscript further. I outline below the specific action taken in response to the comments:
Comment 1: Adding the results of ATX1 expression in the manuscript could strengthen and increase the reliability of the findings.
Response: We agree with this suggestion and in the revised manuscript have included that data in Supplemental Figure S1. The text of the manuscript has accordingly been revised to reflect this: Line 163-164 as well as Lines 350-352 in Discussion.
Comment 2: Many studies suggest that locomotor alterations begin around 15 days, but in the context of the mechanism, selecting 30 days is also considerable.
Response: We agree with the reviewers comment.
Comment 3: Including the ring assay video might enhance the strength of the manuscript.
Response: We have now included sample videos of RING assay of 30 day old Control and SCA1 flies in Supplemental data (Supplemental Video V1 and Supplemental Video V2). This has also been mentioned in the revised text (Lines 157-158).
Comment 4: Overall, the manuscript is well-designed and provides mechanistic insights into Poly-Q diseases.
Response: We certainly appreciate this comment!